# Validity conditions of approximations for a target-mediated drug disposition model: A novel first-order approximation and its comparison to other approximations

Jong Hyuk Byun [1,2], Hye Seon Jeon[3]*, Hwi-yeol Yun[3,4]*, Jae Kyoung Kim[5,6]*

1 Department of Mathematics and Institute of Mathematical Science, Pusan National University, Busan, Republic of Korea, 2 Institute for Future Earth, Pusan National University, Busan, Republic of Korea, 3 College of Pharmacy, Chungnam National University, Daejeon, Republic of Korea, 4 Department of Bio-AI Convergence, Chungnam National University, Daejeon, Republic of Korea, 5 Biomedical Mathematics Group, Pioneer Research Center for Mathematical and Computational Sciences, Institute for Basic Science, Daejeon, Republic of Korea, 6 Department of Mathematical Sciences, KAIST, Daejeon, Republic of Korea

* hyeseon0128@o.cnu.ac.kr (HSJ); hyyun@cnu.ac.kr (H-yY); jaekkim@kaist.ac.kr (JKK)

**Data Availability Statement:** All relevant data are within the manuscript and its Supporting Information files.

## Abstract

Target-mediated drug disposition (TMDD) is a phenomenon characterized by a drug's high-affinity binding to a target molecule, which significantly influences its pharmacokinetic profile within an organism. The comprehensive TMDD model delineates this interaction, yet it may become overly complex and computationally demanding in the absence of specific concentration data for the target or its complexes. Consequently, simplified TMDD models employing quasi-steady state approximations (QSSAs) have been introduced; however, the precise conditions under which these models yield accurate results require further elucidation. Here, we establish the validity of three simplified TMDD models: the Michaelis-Menten model reduced with the standard QSSA (mTMDD), the QSS model reduced with the total QSSA (qTMDD), and a first-order approximation of the total QSSA (pTMDD). Specifically, we find that mTMDD is applicable only when initial drug concentrations substantially exceed total target concentrations, while qTMDD can be used for all drug concentrations. Notably, pTMDD offers a simpler and faster alternative to qTMDD, with broader applicability than mTMDD. These findings are confirmed with antibody-drug conjugate real-world data. Our findings provide a framework for selecting appropriate simplified TMDD models while ensuring accuracy, potentially enhancing drug development and facilitating safer, more personalized treatments.

## Author summary

Target-mediated drug disposition (TMDD) is a phenomenon characterized by the high-affinity binding of a drug to its target molecule. The TMDD model can describe the process to elucidate the binding of the drug to its target and its elimination from the body. However, when target or complex concentrations are not available, simpler models of the TMDD model need to be used to avoid over-parameterization and to improve computational efficiency and analysis. Several simplified TMDD models based on quasi-

**Funding:** J.H.B is supported by the National Research Foundation of Korea (NRF) grant funded by the Korean government (MSIT) (no. RS-2023-00210403) and Learning & Academic research institution for Master's·PhD students, and Postdocs (LAMP) Program of the National Research Foundation of Korea (NRF) grant funded by the Ministry of Education (no. RS-2023-00301938). H.Y. is supported by the National Research Foundation of Korea(NRF) grant funded by the Korea government (MSIT) (no. NRF-2022R1A2C1010929). J.K.K. is supported by the Institute for Basic Science IBS-R029-C3. The funders had no role in study design, data collection and analysis, decision to publish, or preparation of the manuscript.

**Competing interests:** The authors have declared that no competing interests exist.

equilibrium, Michaelis-Menten (MM), or quasi-steady-state (QSS) approximation have been proposed. However, their validity conditions have not been fully investigated. In this study, we derive the validity conditions for the approximations of the TMDD model, providing insights into the appropriate use of simplified models. We also propose a first-order approximation of the QSS model, which is faster than the QSS model and more accurate than the MM model. We also applied the simplified models to antibody-drug conjugate real-world data and obtained the same results. Our work provides clear guidance on the use of the simplified TMDD models, potentially leading to improved drug development and safer, more tailored treatments for patients.

## Introduction

Drugs that bind to their target sites can undergo a nonlinear pharmacokinetic phenomenon that is called "target-mediated drug disposition" (TMDD), first introduced by *Levy* [1]. TMDD occurs when the binding of a drug to its target influences the distribution and elimination of the drug, and is particularly common with biologics, such as monoclonal antibodies [2]. TMDD can be described with a system of ordinary differential equations for a drug, target, and drug-target complex [3]. The TMDD model takes into account the fact that drugs can bind to their target molecules and be eliminated from the body, or they can dissociate from their target molecules and re-enter the circulation. This model is critical to drug development for predicting drug efficacy and safety, selecting drug candidates, and optimizing lead compounds [4–6].

The TMDD model can be simplified to increase computational efficiency, make analysis easier, and provide insights into the behavior of the system [7–9]. Furthermore, it is critical to avoid over-parameterization and to simplify the complexity of the TMDD model [10]. This is particularly important when dealing with relatively sparse clinical data, because of ethical issues. The application of these techniques helps to streamline the model, making it more manageable and interpretable, while ensuring robust and accurate predictions despite limited data availability. Applications generally employ the following processes: parameter reduction, which reduces the number of model parameters; state variable reduction, which reduces the number of state variables in the model; and structural model reduction, which simplifies the structure of the model [11,12].

Previous studies compared the advantages and disadvantages of various TMDD models: the quasi equilibrium (QE) model reduced with quasi-equilibrium, the Michaelis-Menten (MM) model reduced with standard QSSA (sQSSA), and the QSS model reduced with total QSSA (tQSSA) [11,13,14]. The QE model has proven effective under conditions of higher drug concentrations than target concentrations but has not held when the internalization rate of the complex is non-negligible. Subsequently, the MM model has been introduced, streamlining equations and parameters for more efficient outcomes [15]. Although widely applied, the MM model requires significantly higher drug concentrations than the target concentration for validation [10,16]. In addressing these challenges, *Gibianski et al.* introduced the QSS model, which offers improved approximations across various cases [10]. While the validity conditions of the approximations were investigated, the research was performed in limited conditions, such as excessive drug concentration over the target concentration [15,17]. While the condition of excessive drug concentration is common, there are cases when the target concentration is comparable to or exceeds that of the drug, in particular for micro-dosing studies or for drugs having highly non-specific protein binding.

In this study, we derived the validity criteria of approximations of the TMDD model: the MM model (referred to as mTMDD), the QSS model (referred to as qTMDD), and the pTMDD, which is the first-order approximation of the qTMDD model and the first derived in this study. We found that when the criteria were satisfied, the models became accurate. From the criteria analysis, we found that mTMDD is accurate when the initial drug concentrations significantly exceed the target concentrations, as known. On the other hand, the qTMDD is accurate regardless of the relationship between drug and target concentrations. pTMDD is simpler and faster than qTMDD, and it is accurate as long as drug and target concentrations are not similar. These findings are supported by our results from an antibody-drug conjugate real-world data application study. Our findings provide clear guidelines for the use of various TMDD models in the right context.

## Results

### Derivation of the mTMDD, qTMDD, and pTMDD

In this section, the outcomes are described without the detailed derivation of the mathematical models. We recommend reading the materials and methods section in advance for those who wish to review the detailed derivation of the models.

The TMDD model consists of two variables (Table 1). This model can be reduced to a one-variable model, the mTMDD model, by using the sQSSA, yielding the MM model (Table 1). With another model reduction technique, tQSSA [18], another one-variable model, the qTMDD model, can be derived (Table 1). We further simplified the qTMDD model by using the Padé approximant [19], yielding the first-order approximation of the qTMDD model: the pTMDD model (Table 1). While the TMDD model comprises a system of ODEs along with five parameters, all approximations have four parameters.

### The validity criteria for mTMDD, qTMDD, and pTMDD

We derived the validity criteria ($L_m$, $L_q$ and $L_p$) of mTMDD, qTMDD, and pTMDD, which are summarized in Table 1. See Materials and Methods for the detailed derivations. We investigated whether the criteria we derived can predict the accuracy of the models. For this, we utilized the parameter values used in a previous study [10] with various initial drug concentrations: 20 (small), 200 (intermediate), and 2000 (large) units (Table 2).

For these parameters, we calculated $L_m$, $L_q$, and $L_p$ and the relative errors of mTMDD, qTMDD, and pTMDD, respectively (Fig 1A). The values of $L$ and the relative errors are highly correlated, indicating that $L_m$, $L_q$, and $L_p$ can be used as indicators for the accuracy of the

**Table 1. Approximate models compared to the TMDD model and validity criteria.** The TMDD model can be reduced to two differential equations when $k_{int} = k_{deg}$. Other models were formulated using the balance equation and the first-order Taylor expansion (see Materials and Methods).

| | Equations | Validity Criteria ($L$) |
|---|---|---|
| **TMDD** | $\frac{dC}{dt} = -k_{el}C - k_{on}C \cdot R_{tot} + \left(k_{on}C + k_{off}\right)RC$ <br> $\frac{dRC}{dt} = k_{on}C \cdot R_{tot} - \left(k_{on}C + k_{off} + k_{int}\right)RC$ | |
| **mTMDD** | $\frac{dC}{dt} = -k_{el}C - k_{int}\frac{R_{tot} \cdot C}{k_m + C}$ | $L_m = \frac{k_{el}}{k_{on}(C_0 + k_m)} + \frac{R_{tot}}{C_0 + k_m}$ |
| **qTMDD** | $\frac{dC_{tot}}{dt} = -k_{el}C - k_{int}\frac{R_{tot} \cdot C(C_{tot})}{k_m + C(C_{tot})}$ <br> $C(C_{tot}) = \frac{1}{2}\left[(C_{tot} - R_{tot} - k_m) + \sqrt{(C_{tot} - R_{tot} - k_m)^2 + 4k_m C_{tot}}\right]$ | $L_q = \frac{k_{el}}{k_{on}(C_0 + k_m + R_{tot})} + \frac{k_{int}R_{tot}}{k_{on}(C_0 + k_m + R_{tot})^2}$ |
| **pTMDD** | $\frac{dC_{tot}}{dt} = -k_{el}C_{tot} - (k_{int} - k_{el})\frac{R_{tot} \cdot C_{tot}}{k_m + R_{tot} + C_{tot}}$ | $L_p = L_q + \frac{4C_0 R_{tot}}{(C_0 + R_{tot} + k_m)^2}$ |

 

**Table 2. Parameters used for comparisons of the TMDD model, mTMDD, qTMDD, and pTMDD.** The parameter values were obtained from a previous study [10], but $k_{off}$ was changed from 0 to 0.01 in cases 1, 4, and 7 for realistic considerations.

| Cases | $k_{on}$ | $k_{off}$ | $k_{int}$ | $R_{tot}$ | $k_m = \frac{k_{off}+k_{int}}{k_{on}}$ | $k_{el}$ | $\frac{k_{on}R_0}{k_{int}}$ |
|---|---|---|---|---|---|---|---|
| 1 | 0.1 | 0.01 | 2 | 10 | 20 | 1e-5 | 1/2 |
| 2 | 0.1 | 2 | 2 | 10 | 40 | 1e-5 | 1/2 |
| 3 | 0.5 | 0.5 | 2 | 10 | 5 | 1e-5 | 5/2 |
| 4 | 0.1 | 0.01 | 0.2 | 100 | 2 | 1e-5 | 50 |
| 5 | 0.1 | 2 | 0.2 | 100 | 22 | 1e-5 | 50 |
| 6 | 0.5 | 0.5 | 0.2 | 1000 | 1.4 | 1e-5 | 2500 |
| 7 | 0.1 | 0.01 | 0.02 | 1000 | 0.2 | 1e-5 | 5000 |
| 8 | 0.1 | 2 | 0.02 | 1000 | 20.2 | 1e-5 | 5000 |
| 9 | 0.0001 | 0.1 | 3 | 1000 | 31000 | 1e-5 | 1/30 |

simplified models. In particular, when $L_m$ is below 0.1, the relative error of mTMDD is less than 0.1. Furthermore, when $L_q$ and $L_p$ are below 0.6, the relative errors of qTMDD and pTMDD are less than 0.1, respectively.

For mTMDD, when the condition $K_m + C_0 \gg R_{tot}$ is not satisfied (i.e., $L_m$ is not small), it fails to approximate the TMDD model (Fig 1B; Case 4–8). On the other hand, qTMDD is accurate for all of the cases (Fig 1C). This is because $L_q < \frac{k_{el}}{k_{on}(C_0+k_m+R_{tot})} + \frac{k_{int}}{4k_{on}(C_0+k_m)} < \frac{k_{el}}{k_{on}(C_0+k_m+R_{tot})} + \frac{1}{4} \approx \frac{1}{4}$ as long as $k_{el} \ll k_{on}(C_0+k_m)$. In the case of pTMDD, the additional condition $4C_0R_{tot} \ll (R_{tot}+C_0+k_m)^2$ should be met, which holds if $C_0 \ll R_{tot}+k_m$ or $C_0+k_m \gg R_{tot}$ [20]. Thus, pTMDD provides a better approximation than mTMDD when $R_{tot}$ significantly exceeds $C_0$ (Fig 1D; cases 7–8). However, pTMDD becomes inaccurate when $C_0+k_m \approx R_{tot}$ (Fig 1D; cases 4–6). Similar patterns were observed when initial drug concentrations were either 20 (S1 Fig) or 2000 (S2 Fig).

## The simple validity criteria for mTMDD, qTMDD, and pTMDD

Although the validity criteria (Table 1) provide accurate validity conditions for approximations, they contain various parameters, causing inconvenience in their use. Thus, we propose a simpler rule to determine them faster (Table 3). The simplified criteria are derived with the assumption of $k_{el} \ll k_{on}(C_0+k_m)$, which typically holds because $k_{el}$ is small [21–23].

To assess the simple criteria, we adopted a parameter set which was obtained by estimation from clinical PK data using the TMDD model in a previous study [10]. Then, we varied $C_0$ so that $C_0/R_{tot}$ became either 0.1, 1, 10, or 100. For these cases, again all models showed accurate approximations when their validity conditions were satisfied (Fig 2A). When $C_0/R_{tot}$ was small, qTMDD and pTMDD, but not mTMDD, were accurate (Fig 2B and 2C). When $C_0/R_{tot}$ = 1, only qTMDD was accurate (Fig 2B and 2D). Finally, when $C_0/R_{tot}$ was large, all models were accurate (Fig 2F). This result supports the simple criteria (Table 3): qTMDD is generally accurate and pTMDD is more accurate than mTMDD. Typically, $C_0/R_{tot}$ was large, so all approximate models were reliable to use. However, there were cases when $C_0/R_{tot}$ was not large. In particular, this situation was likely to occur at phase 0 of the study, where a micro-dose was actively used. For instance, the micro-dose (100 μg) of warfarin was much lower than its therapeutic dose (5 mg) in Lappin's study [24]. In addition, albumin, the plasma protein for high non-specific binding with warfarin, lowered the concentration of free warfarin concentration. In consequence, the micro-dosing and the presence of albumin have contributed to making the drug concentration even lower than the target protein in the case of warfarin.

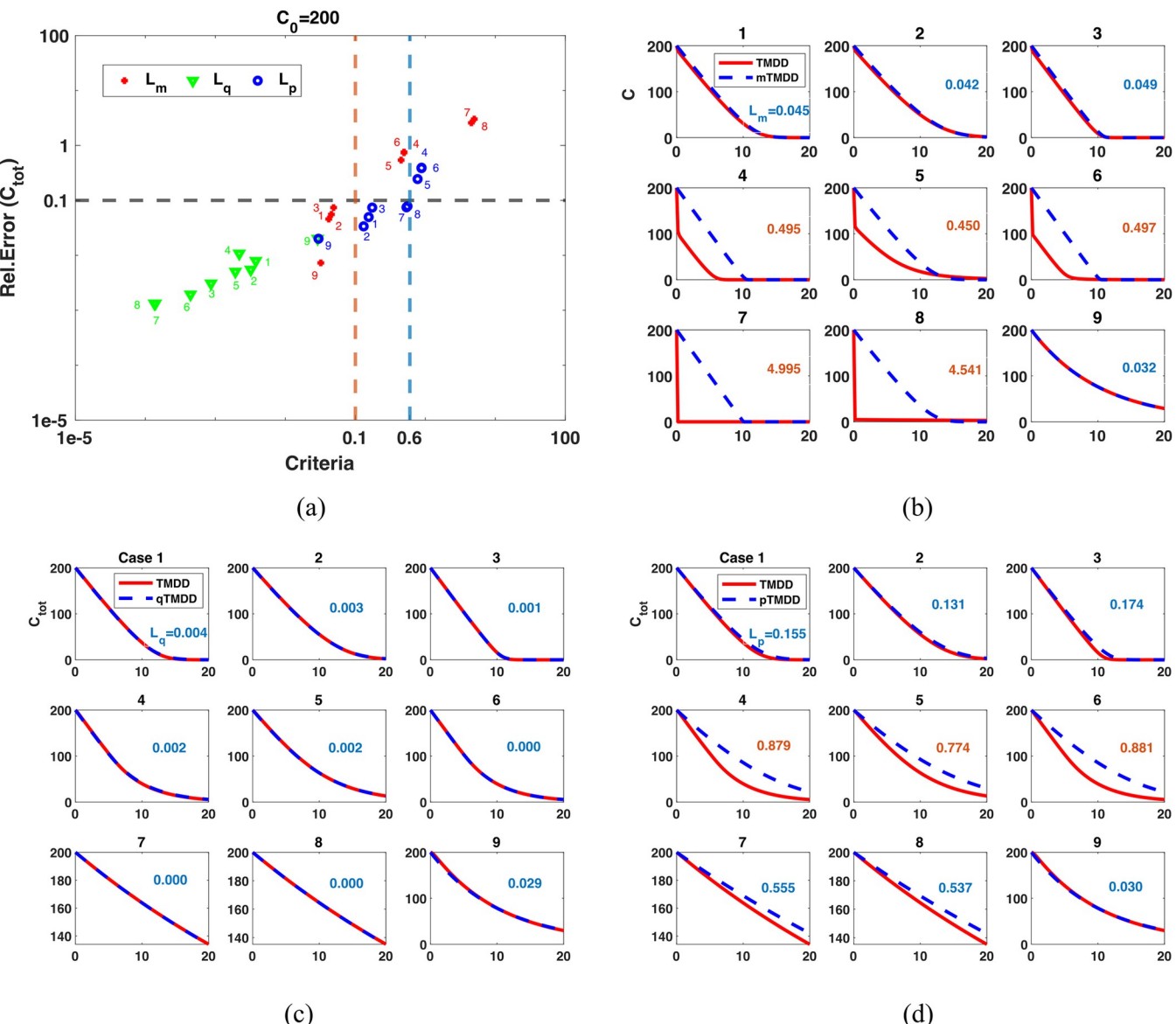

**Fig 1. When $L_m$, $L_q$, and $L_p$ are small, mTMDD, qTMDD, and pTMDD provide accurate approximation for TMDD.** (a) Relative errors of mTMDD, qTMDD, and pTMDD are small when $L_m$, $L_q$, and $L_p$ are small, respectively. The number in the figure represents the case number in Table 2. Here, 200 units of initial drug were used. For 20 and 2000 units of initial drugs, see Figs. S2-S3. (b) mTMDD accurately approximated TMDD when $L_m<0.1$ (blue font) but failed otherwise (red font). The numbers in the figure represent the value of $L_m$. Note that $C$ represents $C_{tot}$ in mTMDD because it assumes $RC$ is negligible. (c) qTMDD accurately approximated TMDD for all cases because $L_q<0.6$. (d) pTMDD accurately approximated TMDD for the total drug when $L_p<0.6$ (blue font) but failed otherwise (red font).

**Table 3. Simple validity criteria.**

|  | mTMDD | qTMDD | pTMDD |
|---|---|---|---|
| $C_0+k_m\gg R_{tot}$ | O | O | O |
| $C_0+k_m\ll R_{tot}$ | X | O | O |
| Otherwise | X | O | X |

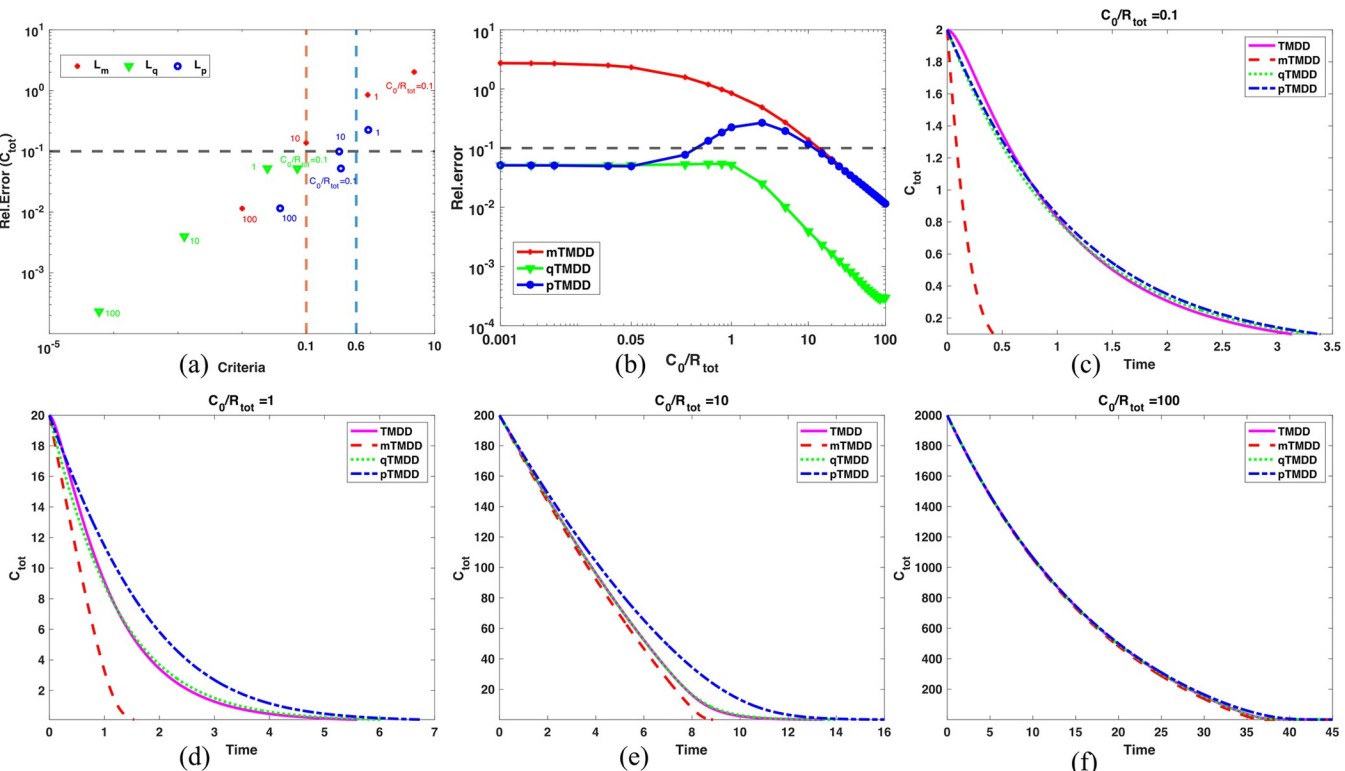

**Fig 2. The accuracy of the models strongly depends on the ratio between drug concentration and total receptor concentration.** (a) Relative errors are plotted against validity criteria for various drug concentrations with the parameter set obtained from [10]. The parameter values of $k_{el}$, $k_{on}$, $k_{off}$, $k_{int}$, and $R_{tot}$ were 0.05, 0.5, 0.1, 1, 20, respectively, so that $k_m = 2.2$. The number shown in the figure represents the value of $C_0/R_{tot}$. (b) The relative errors of the models with respect to $C_0/R_{tot}$. The mTMDD model was accurate only when $C_0/R_{tot}$ was large. The pTMDD model was accurate except for $C_0/R_{tot} \approx 1$. The qTMDD model was accurate regardless of $C_0/R_{tot}$. (c) pTMDD and qTMDD, but not mTMDD were accurate when $C_0/R_{tot} = 0.1$. (d) When $C_0/R_{tot} = 1$, only qTMDD was accurate. (e-f) As $C_0/R_{tot}$ increased, all models became more accurate.

## Application of antibody-drug conjugate real-world data

Next, we applied the approximate models to antibody-drug conjugate real-world data based on a physiologically based mechanistic FcRn model for antibody coupled with TMDD [25]. These PK models with TMDD, mTMDD, qTMDD, and pTMDD were fitted to totals of 640 and 456 time verses plasma concentration data of case 1 for *hIL-1Ra-hyFc* and case 2 for *rhIL-7-hyFc* from randomized clinical trials, respectively.

In the case 1 data, when $C_0/R_{tot}$ (= 22.33) was large, the PK models with mTMDD, qTMDD, and pTMDD provided a similar estimation of the PK parameters with the PK model with TMDD (Table 4), supporting our simple criteria (Table 3). Furthermore, all PK models reasonably explained the observation data according to visual predictive check plots (Fig 3A): all of the model-predicted confidence intervals encompassed 5%, 95th percentile, and median observations, so the approximation models reasonably explained the observation data.

On the other hand, in case 2 data where $C_0/R_{tot}$ (= 0.01) was small, the PK models with qTMDD and pTMDD, but not the mTMDD model, provided a similar estimation with the PK model with TMDD (Table 4). This was consistent with our simple criteria (Table 3). From the result of visual predictive check plots, the PK models with TMDD, qTMDD, and pTMDD suitably explained the observation data (Fig 3B). However, the PK models with the mTMDD model did not explain the observation data since the 95th percentile of the observations was located too low to be adequately encompassed by the model-predicted confidence interval of 95% (Fig 3B).

**Table 4. Summary of the parameters estimated in the antibody-drug conjugate real-world data application study.**

| Model | Case 1 for *hIL-1Ra-hyFc* ($C_0/R_{tot}$ = 22.33) | | | | Case 2 for *rhIL-7-hyFc* ($C_0/R_{tot}$ = 0.01) | | | |
|---|---|---|---|---|---|---|---|---|
| | TMDD | mTMDD | qTMDD | pTMDD | TMDD | mTMDD | qTMDD | pTMDD |
| *OFV* | 1556.40 | 1564.45 | 1556.59 | 1561.90 | 1552.90 | 1611.92 | 1552.90 | 1553.14 |
| *AIC* | 1918.403 | 1926.45 | 1918.59 | 1923.90 | 1796.9 | 1855.92 | 1796.90 | 1797.14 |
| *CL* | 0.210 | 0.209 | 0.207 | 0.199 | 5.26 | 16.8 | 5.23 | 5.27 |
| $K_a$ | 1.22 | 1.17 | 1.21 | 1.23 | 1.22(im), 0.665(sc) | 0.848(im), 0.817(sc) | 1.23(im), 0.697(sc) | 1.24(im), 0.679(sc) |
| *Q* | 0.0285 | 0.0145 | 0.0292 | 0.0279 | 1.21 | 73.5 | 1.21 | 1.3 |
| $V_c$ | 11.4 | 11.6 | 11.2 | 11.3 | 1.96 | 3.96 | 1.97 | 2.07 |
| $V_d$ | 117 | - | - | - | 31.4 | 0.00810 | 28.2 | 35.9 |
| $K_{SS1}$ | - | 5.97 | 233 | 162 | - | 41327.42 | 29824.48 | 29824.52 |
| $K_{SS2}$ | - | 67.8 | 14.1 | 14.7 | - | 29007.90* | | |
| $K_{deg}$ | 0.264* | | | | 0.642* | | | |
| $R_{tot}$ | 2.23* | | | | 1060* | | | |
| $K_{uptake}$ | 0.00952* | | | | 0.00952* | | | |
| $K_{int}$ | 0.206* | | | | 0.642* | | | |
| $K_{recycle}$ | 0.0346 | 0.0104 | 0.0332 | 0.0403 | 0.000607 | 4.68 | 0.000602 | 0.000619 |
| **Interindividual variability ($\omega^2$, variance)** | | | | | | | | |
| $K_a$ | 0.88 | 1.21 | 0.94 | 1.01 | - | - | - | - |
| $K_{recycle}$ | 0.098 | 0.398 | 0.110 | 0.0977 | - | - | - | - |
| $K_{deg}$ | 0.0851 | 0.0798 | 0.0751 | 0.0638 | - | - | - | - |
| *CL* | 0.0497 | 0.0389 | 0.0532 | 0.0553 | - | - | - | - |
| $K_{uptake}$ | 0.886 | 0.927 | 0.813 | 0.827 | - | - | - | - |
| *Q* | - | - | - | - | 0.292 | 13.2 | 0.292 | 0.303 |
| **Additive error (pmol/L)** | 0.188 | 0.237 | 0.184 | 0.185 | 0.00303 | 1.14 | 0.00348 | 0.00486 |
| **Proportional error (ratio)** | 0.115 | 0.0984 | 0.115 | 0.116 | 0.332 | 0.33 | 0.332 | 0.332 |
| **Elapse time (seconds)** | 5345.33 | 403.25 | 1113.53 | 555.99 | 136.09 | 44.35 | 65.26 | 39.11 |

*OFV*: objective function value, *AIC*: Akaike information criterion, *CL*: clearance, $K_a$: absorption rate constant, *Q*: apparent inter-compartment clearance of drug, $V_c$: apparent volume of distribution (central), $V_d$: apparent volume of distribution (peripheral), $K_{SS1}$: equilibrium dissociation rate constant of drug and target binding, $K_{SS2}$: equilibrium dissociation rate constant of drug and FcRn receptor binding, $K_{deg}$: degradation rate of drug at distribution space, $R_{tot}$: total concentration of receptor, $K_{uptake}$: uptake rate of antibody, $K_{int}$: internalization rate constant of FcRn-drug complex, $K_{recycle}$: recycling rate constant from distribution space to a central compartment.

*Those parameters were fixed based on values from the literature.

We also assessed the 'elapsed time,' defined as the time needed to execute estimation using the final model as the starting point. This estimation process was replicated three times using the identical model to calculate an average duration. In every instance, the PK model incorporating TMDD required the longest estimation time. This was followed by the PK model with qTMDD. Conversely, the PK models with pTMDD and mTMDD were more time-efficient. Collectively, these results suggest that pTMDD could serve as a computationally more efficient substitute for qTMDD, without compromising accuracy.

## Discussion

In this study, we investigated the applicability of various TMDD (Target-Mediated Drug Disposition) approximations—mTMDD, qTMDD, and pTMDD—particularly under conditions where the standard TMDD model may be infeasible due to limited data. The mTMDD and qTMDD were derived with the standard QSSA and the total QSSA, respectively. Here, we

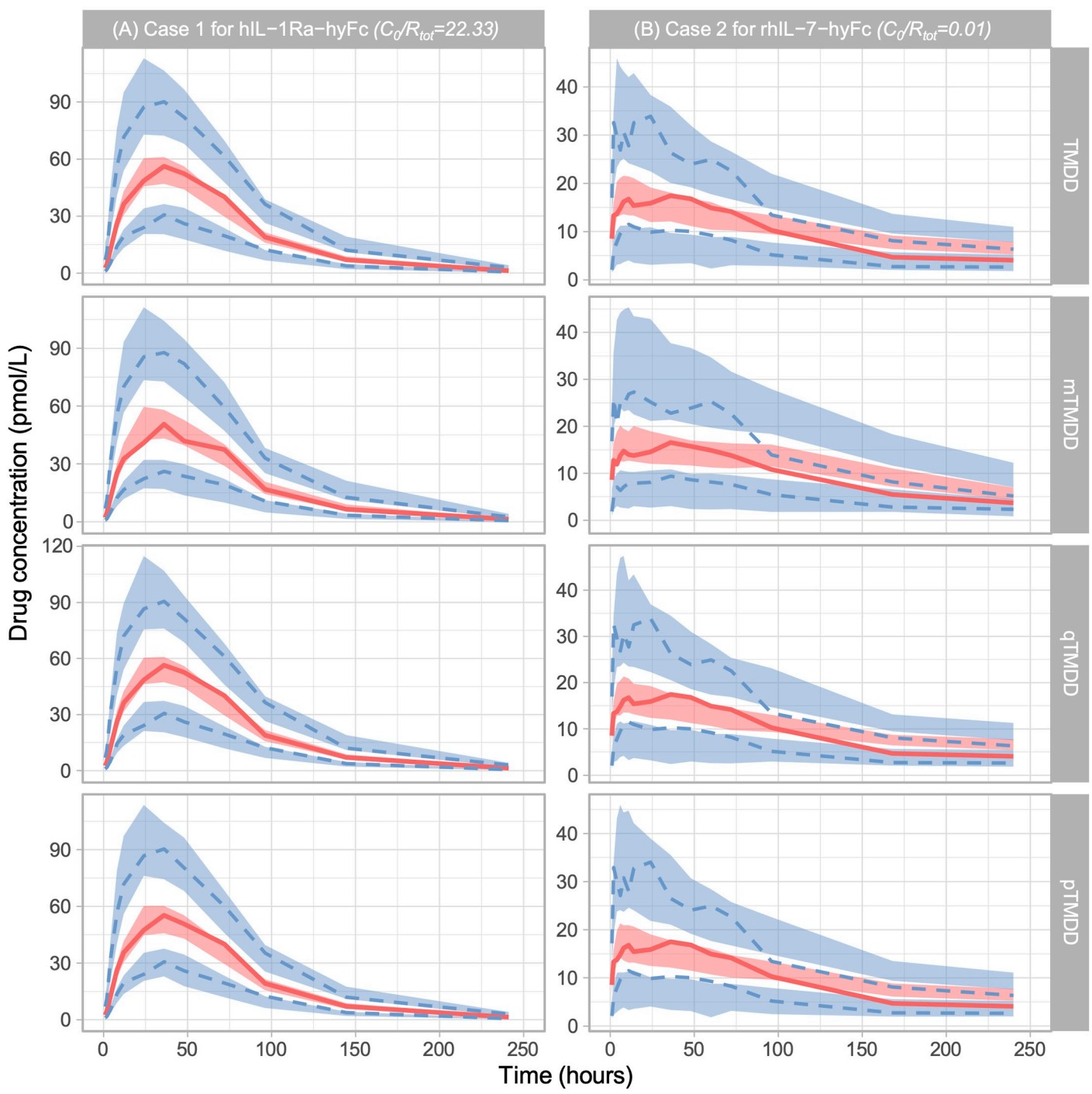

**Fig 3. Visual predictive checks of (A) Case 1 for *hIL-1Ra-hyFc* ($C_0/R_{tot}$ = 22.33) and (B) Case 2 for *rhIL-7-hyFc* ($C_0/R_{tot}$ = 0.01) [25].** The red line indicates the 50th percentile of observations, and the lower and upper blue lines indicate the 5th and 95th percentile of observations, respectively. Blue shades indicate model-predicted confidence intervals of the 5th and 95th percentile, and red shade indicates model-predicted confidence intervals of the 50th percentile. For all PK models of case 1 and the TMDD, qTMDD, and pTMDD PK models of case 2, the model-predicted confidence interval encompassed the 5th percentile, 95th percentile, and median observations. Hence, the approximation models reasonably explained the observation data. For the mTMDD PK model of case 2, the 95th percentile of the observations was low compared to the 95% confidence interval. Therefore, the model cannot be said to predict appropriately.

derived the pTMDD as a first-order Taylor approximation of qTMDD. Specifically, mTMDD was derived with the balanced equation, $k_{on}C \cdot R = (k_{off}+k_{int})RC$, and the total target, $R_{tot} = R+RC$. qTMDD was derived with the conditions of mTMDD and the additional condition of the total drug, $C_{tot} = C+RC$, enabling the derivation of the closed form of $C = C(C_{tot})$. pTMDD was derived with the Taylor expansion of $C = C(C_{tot})$. We established the validity criteria of these approximations, whose accuracies are supported by both simulation and real data analysis. Precisely, mTMDD, qTMDD, and pTMDD were validated by $L_m$, $L_q$, and $L_m$ sufficiently less than one. Additionally, we provided a simpler validity condition for the approximations by comparing $C_0+k_m$ and $R_{tot}$. We found that pharmacokinetic models utilizing mTMDD, qTMDD, and pTMDD yielded estimates consistent with the full TMDD model when drug concentrations exceed the target. Conversely, qTMDD and pTMDD remained accurate even when target concentrations surpassed drug levels, while mTMDD did not align as closely. Notably, pTMDD offered computational efficiency without sacrificing accuracy, making it a viable option in resource-constrained scenarios. Ultimately, our research provides valuable insights into model selection and recommends appropriate approximations for scenarios with limited data availability.

Previous studies suggested that mTMDD is effective when a drug concentration exceeds the target concentration [10,16]. This was rigorously validated in this study. While it was previously known that qTMDD outperforms mTMDD [11], the underlying reasons for this superiority remained unclear. In this study, we found the reason for the accuracy of the qTMDD by deriving its validity criteria. Specifically, the validity criteria of qTMDD holds as long as $k_{el} \ll k_{on}(C_0+k_m)$, which is commonly accepted in pharmacokinetic (PK) studies [21–23]. However, if $k_{el}$ is non-negligible, qTMDD could fail to approximate the TMDD model (S3 Fig).

While mTMDD is derived with sQSSA, the qTMDD is derived with tQSSA. Thus, qTMDD can be used in a wider range of conditions than mTMDD. Similar to this, recently tQSSA has been used to derive alternative equations to predict hepatic drug clearance [26] and drug-drug interactions [27]. These newly derived equations also outperformed the canonical equations based on MM equation derived with standard QSSA. Such outperformance of total QSSA over the standard QSSA has also been reported in modeling and inference of various biological systems [18,26,28–32].

We assumed that the total target concentrations were constant (i.e., $k_{deg} = k_{int}$) in order to prevent the validity criteria complex. This assumption simplified the TMDD model by reducing it to two compartments ($C$ and $RC$). While this assumption is commonly used [11,16,17], developing more flexible criteria without this assumption may offer a more comprehensive assessment of the TMDD model, facilitating better-informed decisions. We also assumed that the drug amount in the peripheral compartment was zero. Without this assumption, the peripheral compartment needs to be incorporated in Eq (1). Future work will investigate a new approach to validity criteria in generalizing the TMDD model considering a non-zero drug amount in the peripheral compartment.

## Materials and methods

### Target-mediated drug disposition (TMDD) model and assumptions

The general TMDD model is as follows [11]:

$$\frac{dA_d}{dt} = -k_a A_d,$$

$$\frac{dC}{dt} = \frac{\ln(t) + k_a A_d}{V} - \left(k_{el} + k_{pt}\right)C - k_{on}C \cdot R + k_{off}RC + k_{tp}\frac{A_T}{V},$$

$$\frac{dA_T}{dt} = k_{pt} C \cdot V - k_{tp} A_T,$$

$$\frac{dR}{dt} = k_{syn} - k_{deg} R - k_{on} C \cdot R + k_{off} RC,$$

$$\frac{dRC}{dt} = k_{on} C \cdot R - \left( k_{int} + k_{off} \right) RC.$$

where, $C$, $R$, $RC$, and $A_T$ represent the free drug, target, complex concentrations in the central (plasma) compartment, and the amount of drug in the peripheral compartment of the tissue, respectively. $A_d$ is the amount of drug in the depot compartment when an oral or subcutaneous dose is administered, and $\ln(t)$ is the infusion rate. Thus, when an IV injection is administered, $A_d = \ln(t) = 0$, and the initial free drug concentration is set to $C(0) = C_0$. The initial values of the target and complex are set as their equilibrium: $R(0) = R_0 = \frac{k_{syn}}{k_{deg}}$ and $RC(0) = 0$.

In addition, the elimination constant $k_{el}$ is defined by $\frac{cl}{V}$, where $V$ and $cl$ represent the systemic volume and clearance, respectively.

To create a simplified model with a more straightforward and concise model structure within the general TMDD framework,

1. The peripheral compartment is not considered.

2. The degeneration rate constant, $k_{deg}$, and the internalization rate constant, $k_{int}$, are the same. If this condition is satisfied, the total target concentration, $R_{tot} = R + RC$, is constant [11,16,17].

With the assumptions above, the TMDD model is simplified as

$$\frac{dC}{dt} = -k_{el} C - k_{on} C \cdot R_{tot} + \left( k_{on} C + k_{off} \right) RC, \tag{1}$$

$$\frac{dRC}{dt} = k_{on} C \cdot R_{tot} - \left( k_{on} C + k_{off} + k_{int} \right) RC. \tag{2}$$

## Model approximations and validity conditions

MM kinetics assume that the binding rate $k_{on} C \cdot R$ is balanced by the sum of dissociation and internalization $(k_{off} + k_{int}) RC$ on the scale of other processes as follows:

$$\frac{C \cdot R}{RC} = \frac{k_{int} + k_{off}}{k_{on}} k_m.$$

Since $R = R_{tot} - RC$, we can get

$$RC = \frac{R_{tot} \cdot C}{k_m + C}.$$

By substituting this to Eq (1) with the balanced equation, $k_{on} C R_{tot} = (k_{on} C + k_{off} + k_{int}) RC$, we obtain mTMDD:

$$\frac{dC}{dt} = -k_{el} C - k_{int} \frac{R_{tot} \cdot C}{k_m + C}.$$

 

This model is valid when the relative change of $C$ should be small during the time when $RC$ approaches equilibrium ($t_c$), which represents initial transient time. In this case, we can estimate $t_c$ by substituting $C \approx C_0$ to Eq (2) as follows:

$$\frac{dRC}{dt} = k_{on}C_0 \cdot R_{tot} - \left(k_{on}C_0 + k_{off} + k_{int}\right)RC.$$

Thus,

$$RC = \frac{R_{tot}C_0}{C_0 + k_m}\left(1 - e^{-k_{on}(C_0 + k_m)t}\right).$$

This indicates that $RC / \frac{R_{tot}C_0}{C_0 + k_m}$ represents the cumulative distribution function of an exponential distribution. Thus, the mean duration of $RC$ is the same as in the exponential distribution with $t_c = 1/[kon(C_0 + km)]$.

Then the validity condition of mTMDD is as follows:

$$\left|\frac{\Delta C}{C_0}\right| = \left|\frac{C(t_c) - C_0}{C_0}\right| \le \frac{1}{C_0}\left|\frac{dC}{dt}\right|_{max} \cdot t_c \le_{C \approx C_0} \frac{1}{C_0}\left(k_{el}C_0 + k_{on}C_0 \cdot R_{tot}\right) \cdot t_c$$

$$= \frac{k_{el}}{k_{on}(k_m + C_0)} + \frac{R_{tot}}{k_m + C_0} \ll 1.$$

We refer to $L_m := \frac{k_{el}}{k_{on}(k_m + C_0)} + \frac{R_{tot}}{k_m + C_0} \ll 1$ as the validity criterion of mTMDD (Table 1). If $k_{el} \ll k_{on}(k_m + C_0)$, this validity criterion of mTMDD can be simplified as $R_{tot} \ll k_m + C_0$ (Table 3).

Next, to derive qTMDD, we consider the total drug concentration, $C_{tot} = C + RC$, whose dynamics are governed by the summation of Eq (1) and Eq (2) as follows:

$$\frac{dC_{tot}}{dt} = -k_{el}C - k_{int}RC. \tag{3}$$

By substituting $RC = C_{tot} - C$ to the balance equation $C \cdot R = k_m RC$, we get $C(R_{tot} - C_{tot} + C) = k_m(C_{tot} - C)$. By solving this quadratic equation, we can express $C$ in terms of $C_{tot}: C(C_{tot}) = \frac{1}{2}\left[(C_{tot} - R_{tot} - k_m) + \sqrt{(C_{tot} - R_{tot} - k_m)^2 + 4k_m C_{tot}}\right]$. By substituting $C$ into Eq (3), we obtain qTMDD:

$$\frac{dC_{tot}}{dt} = -k_{el}C(C_{tot}) - \frac{k_{int} \cdot R_{tot} \cdot C(C_{tot})}{k_m + C(C_{tot})}.$$

qTMDD is valid when the relative change of $C_{tot}$ should be small during the time when $RC$ approaches equilibrium ($t_c$). To estimate $t_c$, we substitute $C_{tot} \approx C_0$ to Eq (2) as follows:

$$\frac{dRC}{dt} =_{C_{tot} \approx C_0} k_{on}C \cdot R - \left(k_{off} + k_{int}\right)RC = k_{on}(C_0 - RC)(R_{tot} - RC) - \left(k_{off} + k_{int}\right)RC$$

$$= k_{on}(C_0 R_{tot} - R_{tot}RC - C_0 RC + RC^2) - k_{on}k_m RC.$$

During this period, $RC$ begins from zero initially and remains relatively small, so we neglect $RC^2$ so that

$$\frac{dRC}{dt} = k_{on}C_0 R_{tot} - k_{on}(R_{tot} + C_0 + k_m)RC.$$

 

As a result, we get $t_c = \frac{1}{k_{on}(R_{tot}+C_0+k_m)}$. Then, the validity condition of qTMDD is as follows:

$$\left|\frac{\Delta C_{tot}}{C_0}\right| = \left|\frac{C_{tot}(t_c) - C_0}{C_0}\right| \leq \frac{1}{C_0}\left|\frac{dC_{tot}}{dt}\right|_{max} \cdot t_c \leq_{C_{tot} \approx C_0} \left|-k_{el} - k_{int}\frac{RC}{C_0}\right| \cdot t_c$$

$$\leq \frac{k_{el}}{k_{on}(C_0 + k_m + R_{tot})} + \frac{k_{int}R_{tot}}{k_{on}(C_0 + k_m + R_{tot})^2} \ll 1,$$

using $RC \leq_{C_{tot} \approx C_0} \frac{C_0 \cdot R_{tot}}{C_0 + k_m + R_{tot}}$. We refer to $L_q := \frac{k_{el}}{k_{on}(C_0 + k_m + R_{tot})} + \frac{k_{int}R_{tot}}{k_{on}(C_0 + k_m + R_{tot})^2} \ll 1$, as the validity criterion of qTMDD (Table 2). If $k_{el} \ll k_{on}(k_m + C_0)$, $L_q < \frac{k_{int}}{4k_{on}(C_0 + k_m)} < \frac{1}{4}$ because $\frac{k_{int}}{4k_{on}(C_0 + k_m)} < \frac{k_{int}}{4k_{on}k_m} = \frac{k_{int}}{4(k_{int}+k_{off})} \leq \frac{1}{4}$. Therefore, if $k_{el} \ll k_{on}(k_m + C_0)$, the qTMDD is generally valid (Table 3).

The pTMDD model is based on the Taylor expansion of $C(C_{tot}) = \frac{1}{2}[(C_{tot} - R_{tot} - k_m) +$

$$\sqrt{(C_{tot} - R_{tot} - k_m)^2 + 4k_m C_{tot}}]$$

$$= C_{tot} - \frac{1}{2}\left[C_{tot} + R_{tot} + k_m - (C_{tot} + R_{tot} + k_m)\sqrt{1 - \frac{4C_{tot}R_{tot}}{(C_{tot} + R_{tot} + k_m)^2}}\right]$$

. The first-order approximation of this equation is as follows:

$$C(C_{tot}) \approx \frac{C_{tot}(C_{tot} + k_m)}{R_{tot} + C_{tot} + k_m},$$

satisfying $r(C_{tot}) := \frac{4C_{tot}R_{tot}}{(C_{tot} + R_{tot} + k_m)^2} \ll 1$. By substituting this to Eq (3), we can obtain pTMDD as follows:

$$\frac{dC_{tot}}{dt} = -k_{el}C_{tot} - (k_{int} - k_{el})\left(C_{tot} - \frac{C_{tot}(C_{tot} + k_m)}{R_{tot} + C_{tot} + k_m}\right)$$

$$= -k_{el}C_{tot} - (k_{int} - k_{el})\frac{C_{tot}R_{tot}}{R_{tot} + C_{tot} + k_m}.$$

The validity condition of pTMDD (Table 2) can be defined as the sum of $L_q$ and $r$, that is,

$$L_p := \frac{k_{el}}{k_{on}(R_{tot} + C_0 + k_m)} + \frac{k_{int}R_{tot}}{k_{on}(R_{tot} + C_0 + k_m)^2} + \frac{4C_0 R_{tot}}{(C_0 + R_{tot} + k_m)^2} \ll 1.$$

If $k_{el} \ll k_{on}(k_m + C_0)$, $L_p < \frac{1}{4} + \frac{4C_0 R_{tot}}{(C_0 + R_{tot} + k_m)^2}$. Furthermore, $\frac{4C_0 R_{tot}}{(C_0 + R_{tot} + k_m)^2} \ll 1$ if $C_0 \ll R_{tot} + k_m$ or $C_0 + k_m \gg R_{tot}$. Therefore, if $k_{el} \ll k_{on}(k_m + C_0)$, the validity criterion of pTMDD can be simplified as $C_0 \ll R_{tot} + k_m$ or $C_0 + k_m \gg R_{tot}$ (Table 3).

## Model application to real-world data of antibody-drug conjugates

Two antibody-drug conjugate clinical trial studies were selected from real case application studies. Case 1 was *hIL-1Ra-hyFc* (human interleukin-1 receptor antagonist components into one antibody-derived fragment crystallizable portion) for the case of large $C_0/R_{tot}$ (= 22.33) and case 2 was *rhIL-7-hyFc* (recombinant human interleukin-7, hybrid Fc-fused) for the case of small $C_0/R_{tot}$ (= 0.01), respectively. Specifically, the observed concentrations data were obtained from clinical trials. See *Ngo. et al.*, [33] and *Lee et al.* [34] for detailed information on the clinical trials.

We applied the three methods (mTMDD, qTMDD, and pTMDD) to the binding of a drug to its target in consideration of biological processes, and other processes were described as

first-order kinetics. In addition, population models that considered inter-individual and residual variability were conducted on the three TMDD methods, respectively. As a general step in the population model development process, inter-individual and residual variability were explored and selected based on numerical (e.g., objective function value, *OFV*) and visual (e.g., goodness of fit, *GoF*) criteria. Parameter optimization for the models was performed using the first-order conditional estimation with interaction (*FOCE-I*) method using NONMEM 7.5 and PsN 5.3.1 software, and model performance was evaluated by model diagnostics criteria and diagnostic plots. The codes and dataset are provided in the supporting information section (S1 Text, S1 and S2 Tables).

## Supporting information

**S1 Text. The NONMEM code of the population pharmacokinetic model with each approximation method implemented.**
(DOCX)

**S1 Table. Dataset for case 1 (hIL-1Ra-hyFc).**
(CSV)

**S2 Table. Dataset for case 2 (rhIL-7-hyFc).**
(CSV)

**S1 Fig. When $L_m$, $L_q$, and $L_p$ are small, mTMDD, qTMDD, and pTMDD provide accurate approximation for TMDD with 20 units of initial drug.** (a) Relative errors of mTMDD, qTMDD, and pTMDD are small when $L_m$, $L_q$, and $L_p$ are small, respectively. The number in the figure represents the case number in Table 2. Here, 20 units of initial drug were used. (b) mTMDD accurately approximated TMDD when $L_m < 0.1$ (blue font) but failed otherwise (red font). The numbers in the figure represent the value of $L_m$. Note that C represents $C_{tot}$ in mTMDD because it assumes *RC* is negligible. (c) qTMDD accurately approximated TMDD for all cases because $L_q < 0.6$. (d) pTMDD accurately approximated TMDD for the total drug when $L_p < 0.6$ (blue font) but failed otherwise (red font).
(TIF)

**S2 Fig. When $L_m$, $L_q$, and $L_p$ are small, mTMDD, qTMDD, and pTMDD provide accurate approximation for TMDD with 2000 units of initial drug.** (a) Relative errors of mTMDD, qTMDD, and pTMDD are small when $L_m$, $L_q$, awpnd $L_p$ are small, respectively. The number in the figure represents the case number in Table 2. Here, 2000 units of initial drug were used. (b) mTMDD accurately approximated TMDD when $L_m < 0.1$ (blue font) but failed otherwise (red font). The numbers in the figure represent the value of $L_m$. Note that C represents $C_{tot}$ in mTMDD because it assumes *RC* is negligible. (c) qTMDD accurately approximated TMDD for all cases because $L_q < 0.6$. (d) pTMDD accurately approximated TMDD for the total drug when $L_p < 0.6$ (blue font) but failed otherwise (red font).
(TIF)

**S3 Fig. When the condition $k_{el} \ll k_{on}(k_m + C_0)$ is not met, qTMDD cannot accurately approximate the TMDD model.** (a) We used the values of parameters from Case 9 of Table 2 except for $R_{tot}$ and $k_{el}$. We used $R_{tot} = 100$ and $k_{el} = 1$. Furthermore, we varied initial drug concentrations ($C_0$) as 0.1, 1, 10 and 100 units so that $C_0/R_{tot}$ changes. $C_0/R_{tot}$ is represented as the numbers in the figure. In all these cases, $k_{el} \ll k_{on}(k_m + C_0)$ is not satisfied because $k_{el} = 1$ and the values of $k_{on}(k_m + C_0)$ are 0.012 ($C_0/R_{tot} = 0.001$), 0.021 (0.01), 0.111 (0.1) and 1.011 (1). As a result, $L_q$ exceeded 0.6 regardless $C_0/R_{tot}$, resulting in relative errors of qTMDD greater than

0.1. (b) Since $L_q>0.6$ (red font), qTMDD fails to approximate TMDD regardless of $C_0/R_{tot}$. (TIF)

## Author Contributions

**Conceptualization:** Jong Hyuk Byun, Jae Kyoung Kim.

**Data curation:** Hye Seon Jeon, Hwi-yeol Yun.

**Formal analysis:** Jong Hyuk Byun.

**Funding acquisition:** Jong Hyuk Byun, Hwi-yeol Yun, Jae Kyoung Kim.

**Investigation:** Jong Hyuk Byun, Hye Seon Jeon, Hwi-yeol Yun, Jae Kyoung Kim.

**Methodology:** Jong Hyuk Byun, Jae Kyoung Kim.

**Project administration:** Jong Hyuk Byun.

**Supervision:** Hwi-yeol Yun, Jae Kyoung Kim.

**Validation:** Hye Seon Jeon, Hwi-yeol Yun.

**Visualization:** Jong Hyuk Byun, Hye Seon Jeon, Hwi-yeol Yun, Jae Kyoung Kim.

**Writing – original draft:** Jong Hyuk Byun, Hye Seon Jeon, Hwi-yeol Yun, Jae Kyoung Kim.

**Writing – review & editing:** Jong Hyuk Byun, Hye Seon Jeon, Hwi-yeol Yun, Jae Kyoung Kim.

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
