## [Decision Letter · Decision Letter 0]

4 Nov 2023

Dear Dr. Byun,

Thank you very much for submitting your manuscript "Validity and Comparative Analysis of Approximations for Target-Mediated Drug Disposition" for consideration at PLOS Computational Biology.

As with all papers reviewed by the journal, your manuscript was reviewed by members of the editorial board and by several independent reviewers. In light of the reviews (below this email), we would like to invite the resubmission of a significantly-revised version that takes into account the reviewers' comments.

The reviewers appreciate the approximation of TMDD provided but raise several points that must be addressed, including data used for fitting and explicitly clarifying details that appear to be glossed over. In addition, the reviewers suggest to apply the method to more relevant case studies to demonstrate the utility.

We cannot make any decision about publication until we have seen the revised manuscript and your response to the reviewers' comments. Your revised manuscript is also likely to be sent to reviewers for further evaluation.

Sincerely,

James Gallo

Academic Editor

PLOS Computational Biology

Stacey Finley

Section Editor

PLOS Computational Biology

Reviewer's Responses to Questions

**Comments to the Authors:**

Reviewer #1: This is an interesting and novel investigation of the important biological equations that are often used in drug development. I do not have any major comments to what was done in the manuscript. However, to make it more useful, it would be helpful to consider a more general case of a two compartment system (all monoclonal antibodies have bi-exponential kinetic) and remove the restriction kint=kdeg. Otherwise, application of the work to real-life cases will be very limited.

With that, I can recommend the manuscript to for the publication.

Reviewer #2: While the work presented in this manuscript have merit and relevance to the development of drugs demonstrating TMDD, the presentation of the work is not lucid, and the author loses the important points being made to the reader in the forest of math. Below are a few suggestions to majorly modify the manuscript.

-From the title itself the author should mention that they are proposing a new approximation of TMDD: something like “A Novel First-Order Approximation of TMDD Model and its Comparison to Other Approximations”. Thus, the reader knows exactly what they are reading.

-While all the math presented in the paper is needed for the experts to verify the work, maybe a lot can be shuttled to supplementary information so author can focus on the performance of their approximation with other approximations.

-The authors need to show the PK data used by them and then quality of fit by each model (by superimposing fits over the observed data). In the absence of such figures, it is hard to assess which model is off by how much. Also, the figures and performance of different approximations is hard to grasp, which may suggest making different/better figures may be of help.

-All the “conditions” in which one model works vs. other is dizzying. Especially because all of them have some symbols less than or higher than other symbols. The author should try to summarize the conditions in a physiological/pharmacological sense so the reader can easily understand the caveats for using each equation.

-In Table 2, some cases have Koff=0. This is impossible and not relevant for TMDD models. Please confine your analysis to parameter ranges that are realistic.

-Eventually it turns out the new approximation proposed by the author is no better than existing qTMDD approximation. Thus, the authors owe the reader an explanation as to why would one still pursue the new approximation, especially since it can be applied to conditions that are lesser than the one allowed by qTMDD. Saying, “qTMDD may have higher run time” is not good enough. Please do some actual estimations and report the run times and prove the point using real data than an expectation.

-Lastly, the author need to read a bit more about the history of TMDD. The first time this was proposed by Levy (PMID: 7924119), and it does not just change metabolism of the drug, it can also change distribution. Maybe the author can benefit from collaboration with a PK/PD Modeling and Simulation expert in the field of protein therapeutics.

Reviewer #3: The manuscript is well written and the methodology is sound. However, the are some critical concerns that needs to be addressed:

1.- Target mediated drug disposition is NOT a phenomenon is which drug metabolism in influence by interaction with target molecules. What is influence is the disposition of the drug, as the term TMDD indicates, and that encompass the distribution and elimination, but not explicitly drug metabolism, specifically for monoclonal antibodies.

2.- The approximation of the TMDD model requires in specific projects depend on the data available. For instance, if available data of free and total drug, as well as free and total target are available, the full TMDD model is identifiable and the quasi-steady-state (QSS), quasi equilibrium (QE) and Michaelis-Menten (MM) are not really needed. However, if only free drug data are available, then the approximations are needed. It would be good to incorporate this concept into the introduction of the manuscript. The simulations should cover different data availability situations, not only when only free drug concentration is available.

3.- The situation where the target concentration exceeds drug concentration is very rare in drug development because the dose of the compounds is increased until the target is covered. Therefore, most of the dose escalation studies are conducted to achieving the maximal drug effect, consequently, the the situation where target concentration exceeds drug concentration is a theoretical concern of limited relevance, given that is very rare to find a situation like this in drug development. As a matter of fact, it would be good that the authors present a case study where this situation become relevant and list the clinical situations where this approach might be helpful. Please note paclitaxel does not exhibit TMDD but the nonlinear PK behavior is due to the liposome formulation.

4.- Please note that the MM approximation of the TMDD has been evaluated considering the dose correction for the initial condition as reported elsewhere (Yan X, Perez-Ruixo JJ, Krzyzanski W. Dose Correction for a Michaelis-Menten Approximation of a Target-Mediated Drug Disposition Model with a Multiple Intravenous Dosing Regimens. AAPS J. 2020 Jan 16;22(2):30.)

**Have the authors made all data and (if applicable) computational code underlying the findings in their manuscript fully available?**

Reviewer #1: Yes

Reviewer #2: Yes

Reviewer #3: None

PLOS authors have the option to publish the peer review history of their article (what does this mean?). If published, this will include your full peer review and any attached files.

Reviewer #1: No

Reviewer #2: **Yes: **Dhaval K. Shah

Reviewer #3: No
---

## [Decision Letter · Decision Letter 1]

11 Mar 2024

Dear Prof Kim,

Thank you very much for submitting your manuscript "Validity Conditions of Approximations for a Target-Mediated Drug Disposition Model: A Novel First-Order Approximation and Its Comparison to Other Approximations" for consideration at PLOS Computational Biology.

As with all papers reviewed by the journal, your manuscript was reviewed by members of the editorial board and by several independent reviewers. In light of the reviews (below this email), we would like to invite the resubmission of a significantly-revised version that takes into account the reviewers' comments.

We cannot make any decision about publication until we have seen the revised manuscript and your response to the reviewers' comments. Your revised manuscript is also likely to be sent to reviewers for further evaluation.

Sincerely,

James Gallo

Academic Editor

PLOS Computational Biology

Stacey Finley

Section Editor

PLOS Computational Biology

Reviewer's Responses to Questions

**Comments to the Authors:**

Reviewer #1: Validity Conditions of 1 Approximations for a Target-Mediated Drug Disposition Model: A Novel First-Order Approximation and Its Comparison to Other Approximations: R1

Jong Hyuk Byun, Hye Seon Jeon, Hwi-yeol Yun and Jae Kyoung Kim

The authors did a lot of work rewriting the manuscript. However, these changes introduced more problems than solved, and I would recommend another major revision. Below are the specific comments that need to be resolved before I can recommend the work for the publication and general recommendation how to organize the manuscript.

1. Terminology: I think mTMDD is the Michaelis-Menten approximation of the TMDD model. This name has a long history and is universally accepted in the pharmacometric community. The authors are free to introduce their own names and notations, but they need to provide the link to common notation and mention that mTMDD is known as Michaelis-Menten approximation as soon as they defined mTMDD. In the same spirit, total QSSA is known as QSS approximation. It is worthwhile to mention that in pharmacometrics, total QSSA is know simply as QSS.

2. Reference Levy [1-5] is misleading. Indeed, Levi introduced the term in [5] but [1-4] are the later works of different authors. Reference [7] is the first that introduced the TMDD equations, and it is strange that it appears only as [7] rather than [2], right after reference on Levi work [5] that should be moved to [1].

3. The entire introduction with the history of TMDD and its approximation is very strange. Rather than provide logical introduction to the author’s work, it reads as an attempt to cite as many papers as possible. This is not helping the reader, but rather does not allow us to follow the author’s logic. Is it really needed for understanding the results?

4. I am not sure that the new style of the paper where all the equations are moved to the end of the work is helpful. It is impossible to follow the results without seeing the formulas. I liked the prior version much more. This version is unreadable. Each statement must be accompanied by the proof, and it is impossible to follow the proof if it is separated from the results. Please return the materials and method section to their proper place after the introduction.

5. “Materials and methods” section is better written and easier to follow. One place that is not clear is the derivation of mTMDD (lines 273-274). It seems that (2) was used as equality in derivation of mMDD from (1). Please show intermediate steps and explain the assumptions in more detail.

6. Line 281: what is mean duration of RC, how it is defined? Looks like it is similar to half-life, but without log(2).

7. The section that describes application to the real data is interesting but not clear. Where does this data come from? Were the observed data available to the authors or these were data simulated from the model? The authors should show the equations of the models fit to the data. Table 4 has parameter estimates for the linear part of the equations, but other parameters also should be shown (kon, koff, Rtot, kint, KM, kdeg, etc.). Was this population model (with inter-subject variability considered)? How many subjects were there, how well variances were estimated?

Overall, the manuscript has a nice idea: to introduce the new approximation as a Tailor expansion of the QSS. It has merit in some cases. Another interesting idea is to derive applicability conditions of each approximation via some regular procedure (that is not explained in sufficient detail, and thus the reader unfamiliar with the argument cannot follow it). Applications are of interest to demonstrate the place of the new approximation in the family of different simplifications of the TMDD model. However, the manuscript in the current form is unreadable, loaded with too many references in no logical order, not well organized. My advice to the authors would be to reorganize the text in some logical order: a very brief history of the field, then TMDD equations, idea how to derive approximations and how to derive conditions for those, then the derivation of the conditions, and new approximation, and then applications (with clear statement of each problem, equations of the models, equations for the approximations, Nonmem code in supplemental materials, etc.). A short discussion may follow to outline the advantages of the new approximation, summary of conditions, etc. Discussion should mention that while the new approximation is valid for a more general case of non-constat Rtot and two-compartment model, conditions are tailored to simpler case of 1-compartment model with constant Rtot.

Without significant improvement of the manuscript text, this interesting work cannot be comprehended by the readers, and thus, cannot be published.

Reviewer #2: The authors have addressed all the comments in a satisfactory manner and have also changed the title and significantly revised the manuscript as per reviewers' suggestions. The manuscript is now in a better shape and the reviewer has not further changes to suggest.

**Have the authors made all data and (if applicable) computational code underlying the findings in their manuscript fully available?**

Reviewer #1: **No: **More detailed description of the data and Nonmem code of the models used for application should be included in the supplemental materials.

Reviewer #2: Yes

PLOS authors have the option to publish the peer review history of their article (what does this mean?). If published, this will include your full peer review and any attached files.

Reviewer #1: No

Reviewer #2: No
---

## [Decision Letter · Decision Letter 2]

10 Apr 2024

Dear Prof Kim,

We are pleased to inform you that your manuscript 'Validity Conditions of Approximations for a Target-Mediated Drug Disposition Model: A Novel First-Order Approximation and Its Comparison to Other Approximations' has been provisionally accepted for publication in PLOS Computational Biology.

Best regards,

James Gallo

Academic Editor

PLOS Computational Biology

Stacey Finley

Section Editor

PLOS Computational Biology

Reviewer's Responses to Questions

**Comments to the Authors:**

Reviewer #1: The authors successfully accounted for all comments. I recommend to accept the manuscript for the publication

**Have the authors made all data and (if applicable) computational code underlying the findings in their manuscript fully available?**

Reviewer #1: Yes

PLOS authors have the option to publish the peer review history of their article (what does this mean?). If published, this will include your full peer review and any attached files.

Reviewer #1: No

---

## [Editor Report · Acceptance letter]

16 Apr 2024

PCOMPBIOL-D-23-01282R2 

Validity Conditions of Approximations for a Target-Mediated Drug Disposition Model: A Novel First-Order Approximation and Its Comparison to Other Approximations

Dear Dr Kim,

I am pleased to inform you that your manuscript has been formally accepted for publication in PLOS Computational Biology. Your manuscript is now with our production department and you will be notified of the publication date in due course.

With kind regards,

Zsofia Freund
